# Study on the Influence Factors of the Dynamic Property of the Polyurethane Mixture with Dense Gradation

Haisheng Zhao [1,2], Shiping Cui [1], Zhen Li [3], Shaobin Wang [4], Lin Wang [1], Wensheng Zhang [5], Chunhua Su [1], Peiyu Zhang [1] and Shijie Ma [1,*]

[1] Key Laboratory of Highway Maintain Technology Ministry of Communication, Jinan 250102, China; zhaohaisheng@sdjtky.cn (H.Z.); cuishiping@sdjtky.cn (S.C.); wanglin@sdjtky.cn (L.W.); suchunhua@sdjtky.cn (C.S.); zhangpeiyu@sdjtky.cn (P.Z.)

[2] School of Highway, Chang'an University, Xi'an 710064, China

[3] Shandong Tongda Luqiao Planning & Design Co., Ltd., Yantai 264119, China; lizhenokey@163.com

[4] Shandong Transportation Department, Jinan 250002, China; tjbzy2018@163.com

[5] Wanhua Chemical Group Co., Ltd., Yantai 265599, China; wszhangb@whchem.com

* Correspondence: mashijie@sdjtky.cn; Tel.: +86-186-6016-3082

**Abstract:** Similar to the asphalt mixture, the polyurethane (PU) mixture's performance and characteristics are dependent on many variables. In this study, six variables, including aggregate gradation (limestone and basalt), aggregate type, PU type, PU content, and curing condition, and several parameter analyzing methods were chosen to determine the effect of variables on the dynamic property, rheological property, and rutting resistance of the PU mixture. The limestone aggregate gradation exhibited a substantial effect on the dynamic property, rheological property, and rutting resistance of the PU mixture; the basalt aggregate gradation exhibited significant influence on the dynamic property and rutting resistance, but a moderate effect on the rheological property. The aggregate type could influence the rheological property and rutting resistance. The slow curing speed of the PU binder decreased the dynamic modulus and rutting resistance but did not influence the phase angle. The rise in PU binder content would only improve the PU mixture's resistance to rutting. The curing condition and color additive had no impact on the PU mixture's properties. The generalized logistic sigmoidal (GLS) and Christensen Anderson and Marasteanu model (CAM) models could precisely predict the dynamic modulus and phase angle respectively disregarding the PU mixture features. PUM-10/B exhibited the greatest rutting resistance. The findings will aid in comprehending the properties and influencing factors of the PU mixture as well as in designing the desired mixture.

**Keywords:** PU mixture; dynamic property; rheological property; rutting resistance; black space diagram

## 1. Introduction

The asphalt mixture is a complex mixture with linear viscoelastic and rheological characteristics. The mechanical analysis of the asphalt mixture and asphalt pavement design needs effective and accurate parameter characterization to comprehend the behavior and strain response of the asphalt pavement, whereas before analyzing the asphalt mixture strain response [1–4] and material selection of the asphalt pavement design [5,6], it is essential to understand the rheological property of the asphalt mixture. As the asphalt mixture is a viscoelastic material, its mechanical property is dependent on temperature and loading frequency [7,8]. To characterize the viscoelastic property of the asphalt mixture [2,5,6] in the asphalt pavement structural design, researchers have developed many test methods. Since the 1950s [5,6,9–11], the parameters of dynamic modulus and corresponding phase angle have been used to characterize the fundamental viscoelastic, rheological, and time–temperature-dependence properties of the asphalt mixture in undamaged states. The dynamic modulus of the asphalt mixture has become one of the most essential parameters for flexible pavement design and pavement structural evaluation [12,13]. For instance, the

dynamic modulus is one primary input parameter for the mechanistic–empirical pavement design guide (MEPDG) which is based on the mechanical experience method and uses the dynamic modulus master curve to estimate the pavement's structural capacity [14]. The dynamic modulus is also used to estimate mechanical response, performance prediction, and deformation estimation, such as fatigue cracking, fatigue, and rutting models [15,16]. The dynamic modulus, which was reported in NCHRP report 513, was correlated well with the field deformation behavior and used to evaluate the performance of the asphalt mixture in the Superpave design system [17].

The dynamic modulus test is required to be conducted at a specific temperature and loading frequency, and would be impacted by the reaction of the asphalt mixture under a given set of test conditions; therefore, the dynamic modulus is not independent. The asphalt mixture consists of air void, aggregate, and asphalt binder, the performance of the asphalt mixture would be impacted by those components. Numerous studies were performed to determine the impact of the asphalt mixture components on the dynamic modulus data. Nguyen et al. [18] showed that the additive type, aging, temperature, and loading frequency had a significant influence on the dynamic modulus at a significance of 95%. Harvey et al. [19–21] suggested that the air void or the degree of compaction could considerably affect the stiffness of the asphalt mixture. The resilient modulus of the asphalt mixture may be affected by the asphalt mixture's modifier, according to research [22]. The type of asphalt was proven to affect the dynamic modulus and phase angle master curves of the asphalt mixture [23]. Wang et al. [24] selected eight types of asphalt mixture to ascertain the factors that influence the uniaxial dynamic modulus. The dynamic modulus was found to depend on variables such as the aggregate gradation, asphalt binder type, aging degree, and volume parameters [25–32]. Islam et al. [33] confirmed the aforementioned conclusion that the dynamic modulus would increase as the effective asphalt content, air void, and void-filled asphalt content increased, while it would decrease as voids–mineral aggregate and asphalt content increased. Zhang et al. [34] discovered that the porous asphalt (PA) mixture's dynamic modulus decreased as the air voids increased, with and without moisture conditioning. El-Hakim et al. [35] found that the modified asphalt mixture exhibited greater dynamic modulus values after freeze–thaw moisture conditioning than the traditional asphalt mixture; the modification of the asphalt would inherit the stiffness degradation of the asphalt mixture due to moisture. Zhu et al. [36] analyzed three types of asphalt modifiers and concluded that the addition of modifiers would all improve the asphalt mixture's dynamic modulus. As air voids increased, the dynamic modulus degradation of the asphalt mixtures subjected to various freeze–thaw moisture conditioning times would increase [10]. The factors impacting the dynamic modulus of the high-modulus asphalt mixture were studied [33]. The asphalt binder properties could significantly affect the performance and the inherent uncertainty of the asphalt mixture [37]. The study [38] investigated the effect of factors other than temperature and loading frequency on the asphalt mixture's dynamic modulus. The effect of basalt fiber content on the dynamic modulus and rutting factors of the SBS-modified asphalt mixture were investigated, and the optimal fiber content with the best performance was proposed [39]. The effect of oxidative aging of the asphalt binder on the dynamic modulus and viscoelastic properties of the asphalt mixture has been thoroughly studied [40–42]. The influence of aggregate gradation and air voids on the dynamic modulus of the asphalt mixture was analyzed [43]. The asphalt mixture's properties, e.g., aging degree, recycled asphalt mixture material, regenerating agent and dose, and asphalt binder type, were proven to have a considerable impact on its dynamic modulus [44]. The commonly accepted prediction model of Hirsch is the function of binder modulus (G*), voids in the mineral aggregate (VMA), and voids filled with asphalt (VFA). The MEPDG employed the dynamic modulus prediction equation for flexible pavement design and performance prediction of the asphalt mixture. The prediction equation comprises aggregate gradation, effective binder content, binder properties, air voids, temperature, and loading rate. Those findings demonstrate that the dynamic modulus of the asphalt mixture is affected by the aforementioned variables.

Similar to the asphalt mixture, the PU mixture is a kind of mixture consisting of aggregate, mineral powder, and PU binder. Several factors, such as asphalt property, asphalt content, aggregate type, and aggregate gradation, would influence the asphalt mixture performance, as determined by many studies on asphalt mixtures. The dynamic and rheological properties play a crucial role in characterizing the asphalt mixture for material evaluation and flexible pavement design, and are widely utilized by scholars and engineers in research and project designing. Numerous studies were performed to identify the influencing factors of the dynamic and rheological properties of the asphalt mixture to obtain the optimal asphalt mixture with the desired deformation damage resistance. Consequently, the performance of the PU mixture is also dependent on various factors. Urgent research into the influencing factors of the dynamic and rheological properties of the PU mixture, which could help in characterizing the PU mixture, is required. Before applying the PU mixture, it should be thoroughly studied and the factors which may affect its performance should be analyzed. In this study, six variables, including aggregate type (limestone and basalt), gradation, PU type, PU content, and curing condition, were chosen to determine their impact on the dynamic and rheological properties of the PU mixture. Using the dynamic modulus and phase angle master curve, master curve fitting accuracy, black space diagram, and stiffness parameter, it was determined whether the factors had a significant effect on the dynamic and rheological properties of the PU mixture.

## 2. Materials and Methods

### 2.1. Material and Gradation

There were four mixture gradation types, including PUM-20, PUM-16, PUM-13, and PUM-10, chosen for analysis in this study. Two aggregates, including limestone and basalt, were utilized in this paper. The optimal gradation and binder content of the asphalt mixture was determined by the Marshall design method. Figure 1 depicts the selected gradations with various aggregate types.

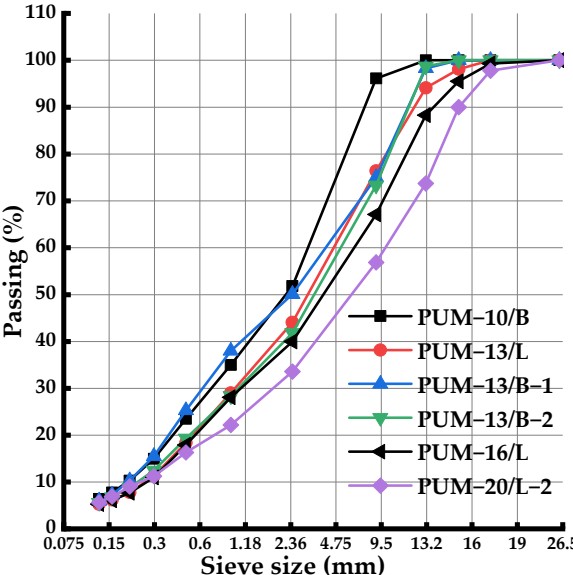

**Figure 1.** The chosen dense gradations with different aggregate types.

In Figure 1, the letter B represents the basalt aggregate, while L represents the limestone. This paper employed three kinds of PU binder: the traditional curing speed PU binder (named T), the slow curing speed PU binder (named S), and the traditional curing speed PU binder with color additive 6179H (named H). The PU binder replaced the same content of the asphalt binder to fabricate the specimens. The PU binders were supplied by Wanhua Chemical Group Co., Ltd. (Yantai, China). Table 1 shows the indexes of the PU binders. Table 2 lists all the PU mixtures with distinct features.

**Table 1.** The index of PU with different cure speeds.

| Index | Test Result | |
|---|---|---|
| | Traditional Cure Speed | Slow Cure Speed |
| Viscosity (25 °C) (MPa·s) | 1707 | 1691 |
| Dry time (30 °C, 90% RH) (min) | 70 | 83 |
| Tensile strength (MPa) | 24.5 | 29.4 |
| Breaking elongation (%) | 212 | 516 |

**Table 2.** The PU mixtures with various features.

| Number | Type | Aggregate Type | PU Content (%) | PU Type |
|---|---|---|---|---|
| 1 | PUM-10/B | Basalt | 5.2 | Traditional cure speed |
| 2 | PUM-13/B-2 | Basalt | 5.2 | Traditional cure speed |
| 3 | PUM-13/L | Limestone | 5.2 | Traditional cure speed |
| 4 | PUM-16/L | Limestone | 4.9 | Traditional cure speed |
| 5 | PUM-20/L | Limestone | 4.9 | Traditional cure speed |
| 6 | PUM-13/B-1/5.0 | Basalt | 5 | Traditional cure speed |
| 7 | PUM-13/B-1/5.6 | Basalt | 5.6 | Traditional cure speed |
| 8 | PUM-13/B-1/5.3(T) | Basalt | 5.3 | Traditional cure speed |
| 9 | PUM-13/B-1/S | Basalt | 5.3 | Slow cure speed |
| 10 | PUM-13/B-2/H | Basalt | 5.2 | Traditional PU binder with color additive 6179H |
| 11 | PUM-20/L/50% | Limestone | 4.4 | Curing condition of 50% RH |
| 12 | PUM-20/L/70% | Limestone | 4.4 | Curing condition of 70% RH |

In Table 2, B-1 and B-1 represent different basalt gradations. The gradation of PUM-20/L/50% and PUM-20/L/70% correspond to PUM-20/L-2 in Figure 1, whereas the gradation of PUM-20/L corresponds to PUM-20/L in Figure 1.

### 2.2. Specimen Fabricating

The PU mixture specimens were compacted 100 times with the Superpave gyratory compactor (SGC) (Pine Test Equipment, Inc., Grove City, OH, USA). After the aggregate gradation and optimal asphalt content were determined using the Marshall design method, the PU binder would replace the asphalt with the same content. The process of specimen preparation of the PU mixture is as follows: (a) the aggregates and mineral filler should be dried with the blast oven at 170 °C for four hours; (b) after the aggregates cool to room temperature, the aggregates, filler and PU binder should be mixed without reheating for 90 s; (c) the loose mixed mixture should remain in the room for about 1.5 h before compaction; (d) the specimens should be stored at 35 °C and 50% RH for 5 days after compacting; (e) the specimens should be sewed into the size of 100 mm in diameter and 150 mm in height.

### 2.3. Dynamic Modulus Test

The dynamic modulus test adhered to AASHTO: TP-79 (2010) specification by using the Asphalt Mixture Performance Tester (AMPT, IPC Global, Melbourne, Austilia). The loading is in the form of the sine wave, the control mode is the strain mode, and the strain of the specimen during the test would be within 75–125 uε to ensure that the specimen will not produce damage or elastic deformation. The test temperature in this study ranged from 5 to 55 °C with a 10 °C increment. The loading frequency was chosen as 25, 20, 10, 5, 2, 1, 0.5, 0.2, and 0.1 Hz. The average values from two replicates of each mixture were tested and were used for further analysis.

# 3. Results

## 3.1. Comparing Master Curve Fitting Results

### 3.1.1. The GLS Model Fitting Results for Dynamic Modulus

Figure 2 depicts the dynamic modulus master curve fitting results for the PU mixture by using the GLS model and WLF shift factor equation. The fitting results, including the master curves and shift factors, were sorted into five groups for analyzing various factors, i.g., aggregate gradation with different aggregate types, aggregate type, PU type, PU content, and curing condition.

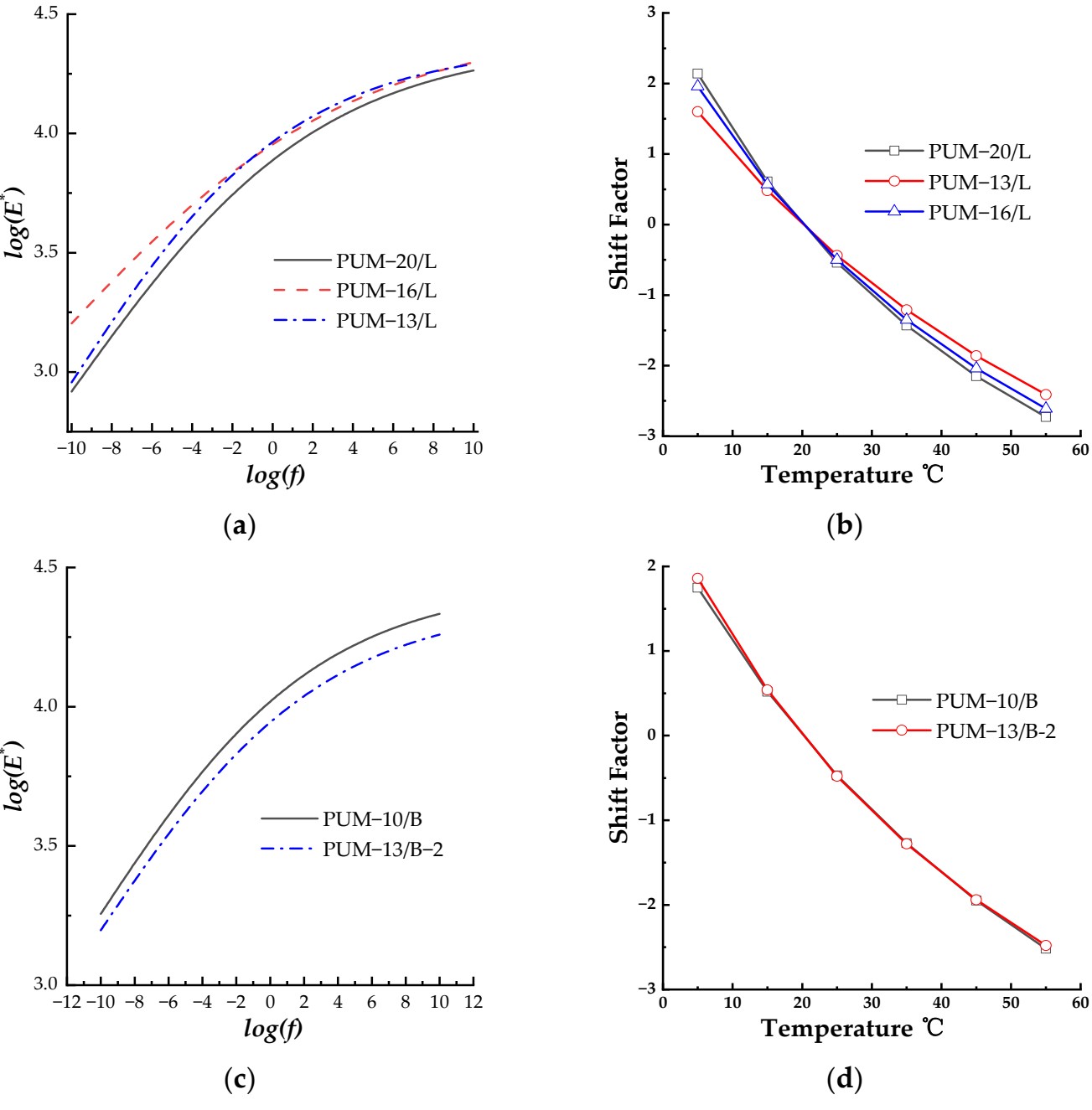

**Figure 2.** *Cont.*

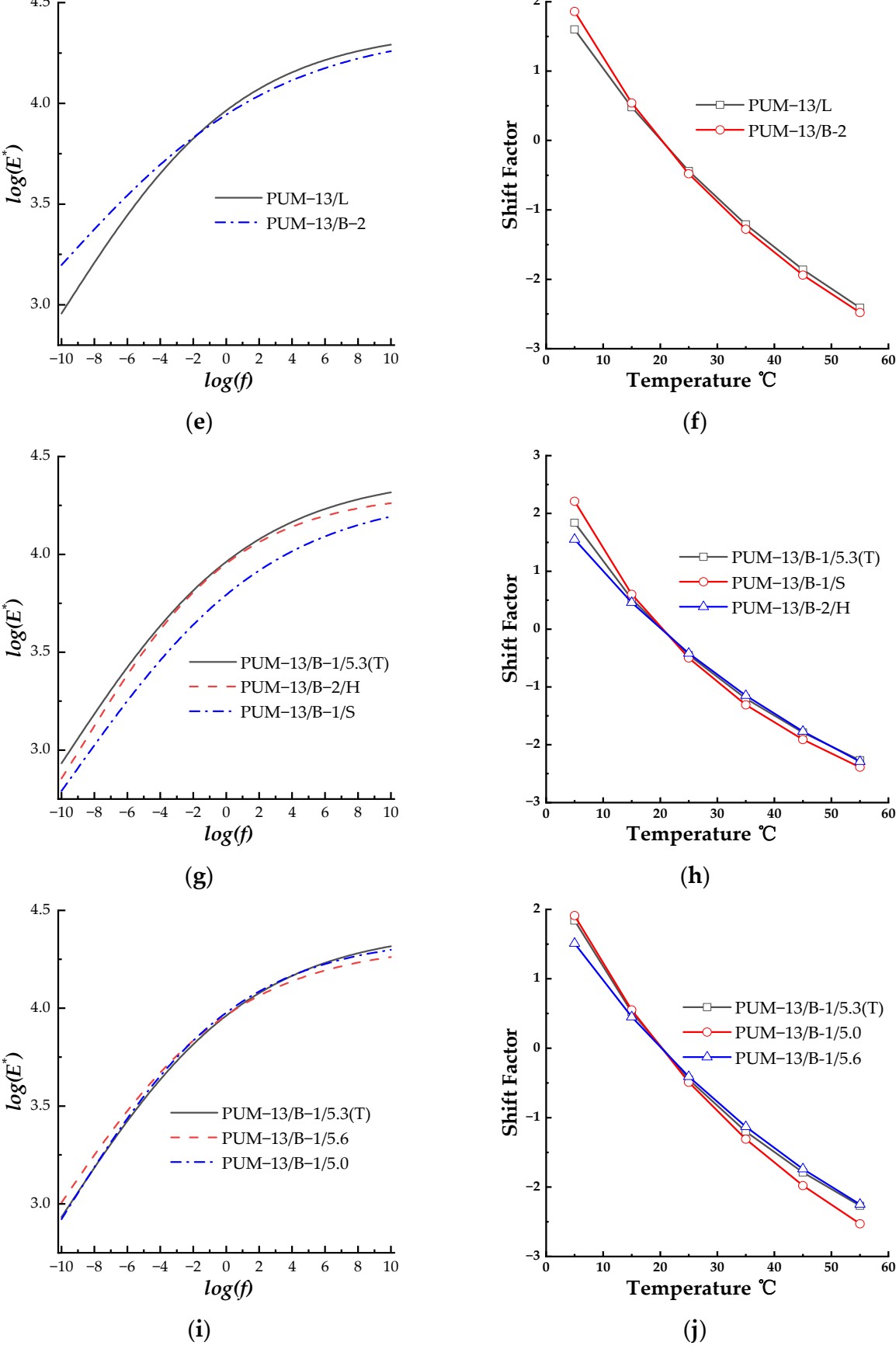

**Figure 2.** *Cont.*

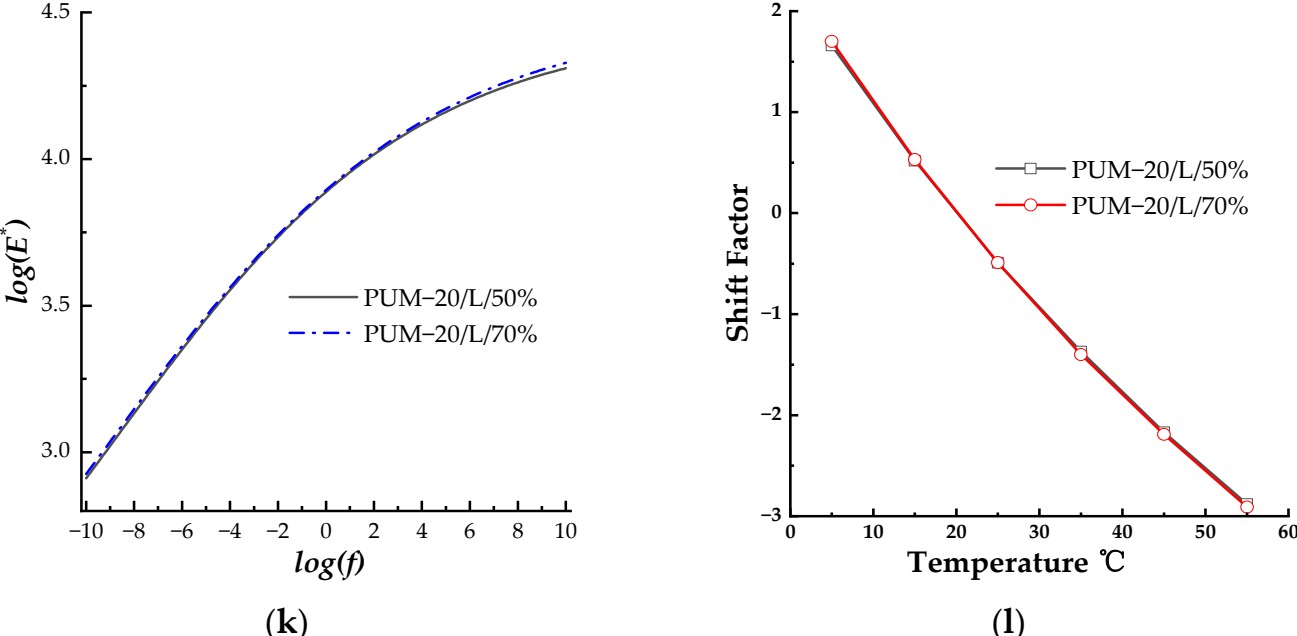

**Figure 2.** The dynamic modulus master curve fitting results found by using the GLS model and WLF shift factor equation. (**a**) Master curves under different limestone aggregate gradations; (**b**) shift factors under different limestone aggregate gradations; (**c**) master curves under different basalt aggregate gradations; (**d**) shift factors under different basalt aggregate gradations; (**e**) master curves under different aggregate types; (**f**) shift factors under different aggregate types; (**g**) master curves under different PU types; (**h**) shift factors under different PU types; (**i**) maser curves under different PU contents; (**j**) shift factors under different PU contents; (**k**) master curves under different curing conditions; (**l**) shift factors under different curing conditions.

### 3.1.2. The CAM Model Fitting Results for Phase Angle

The CAM model and Kaelble shift factor equation were combined to construct the phase angle master curve of the PU mixtures, and the fitting results, including the master curves and shift factors, are shown in Figure 3 for comparing various influencing factors, i.e., aggregate gradation with different aggregate type, aggregate type, PU type, PU content, and curing condition.

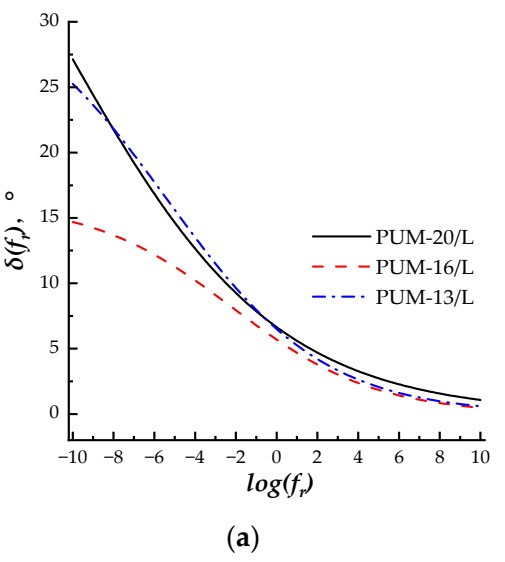

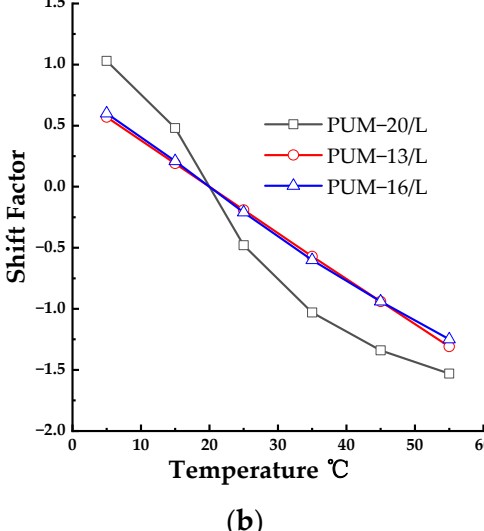

**Figure 3.** *Cont.*

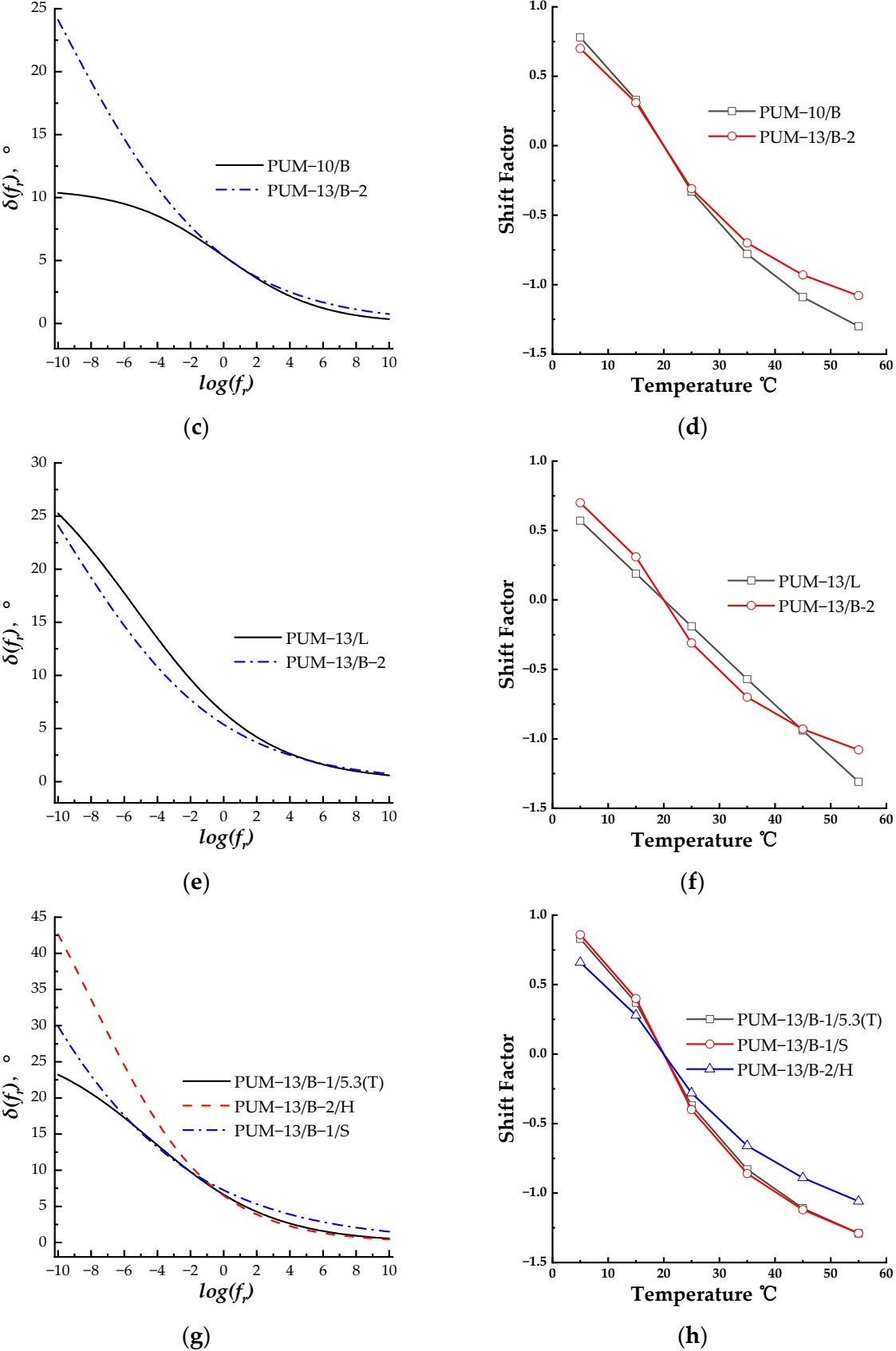

**Figure 3.** *Cont.*

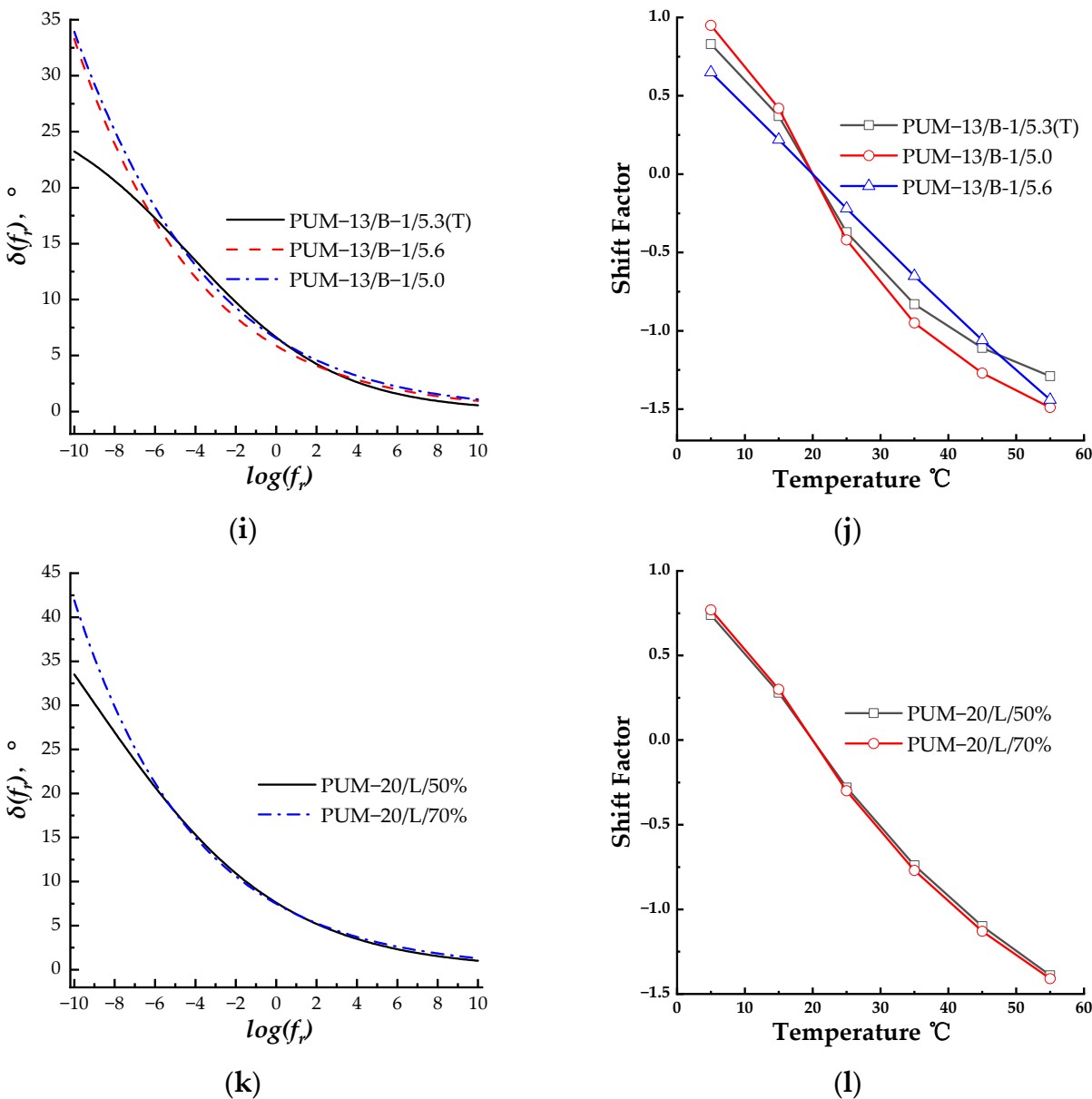

**Figure 3.** The phase angle master curve fitting results found by using the CAM model and WLF shift factor equation. (**a**) Master curves under different limestone aggregate gradations; (**b**) shift factors under different limestone aggregate gradations; (**c**) master curves under different basalt aggregate gradations; (**d**) shift factors under different basalt aggregate gradations; (**e**) master curves under different aggregate types; (**f**) shift factors under different aggregate types; (**g**) master curves under different PU types; (**h**) shift factors under different PU types; (**i**) maser curves under different PU contents; (**j**) shift factors under different PU contents; (**k**) master curves under different curing conditions; (**l**) shift factors under different curing conditions.

### 3.2. Comparing the Accuracy of Different Models

In this section, the measured data and the predicted data for the PU mixture with dense gradation are compared to evaluate the prediction accuracy of various models. Figure 4 displays the comparison results of the SLS and SCM models for the dynamic modulus, while Tables 3 and 4 provide the linear correlation analysis results.

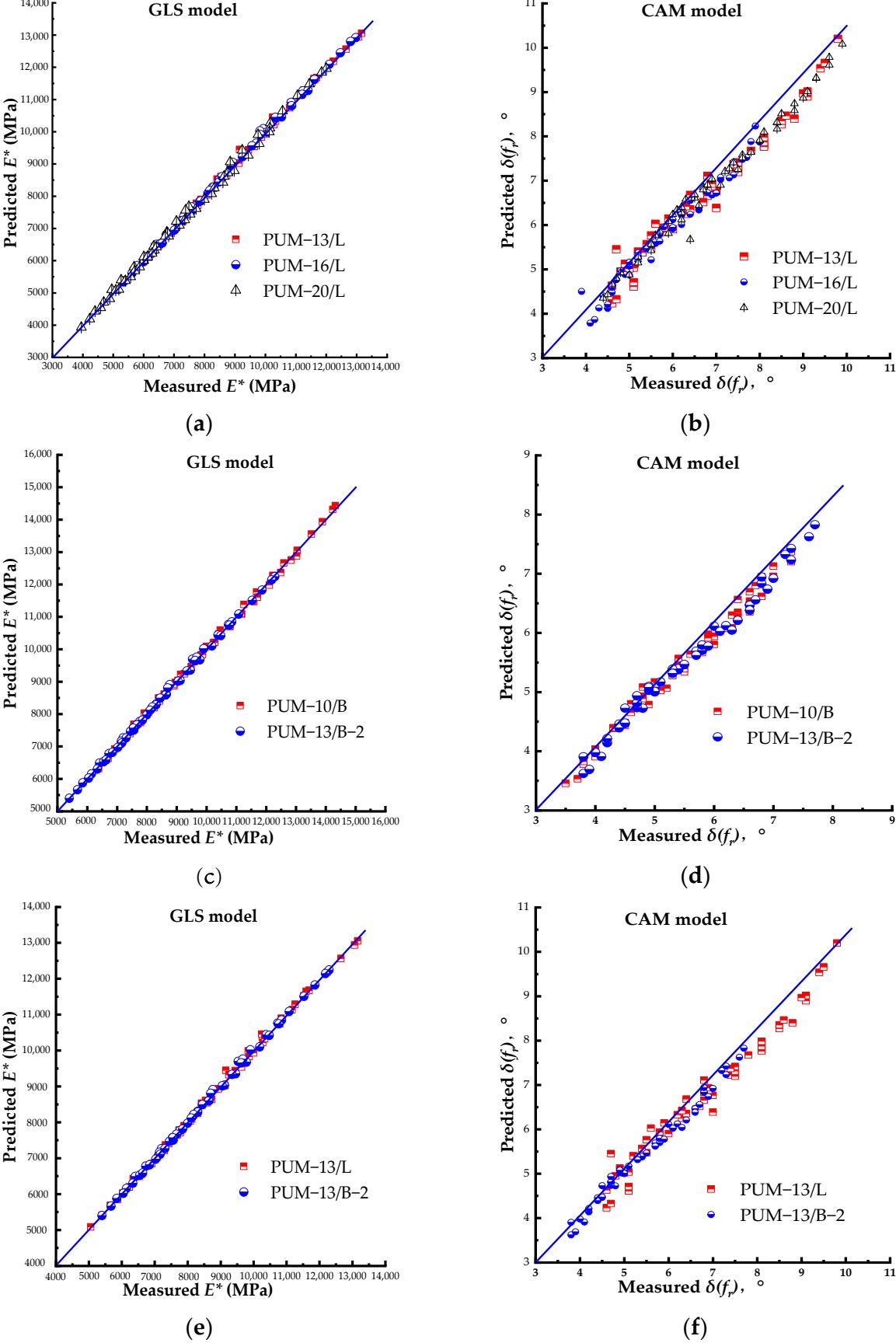

**Figure 4.** *Cont.*

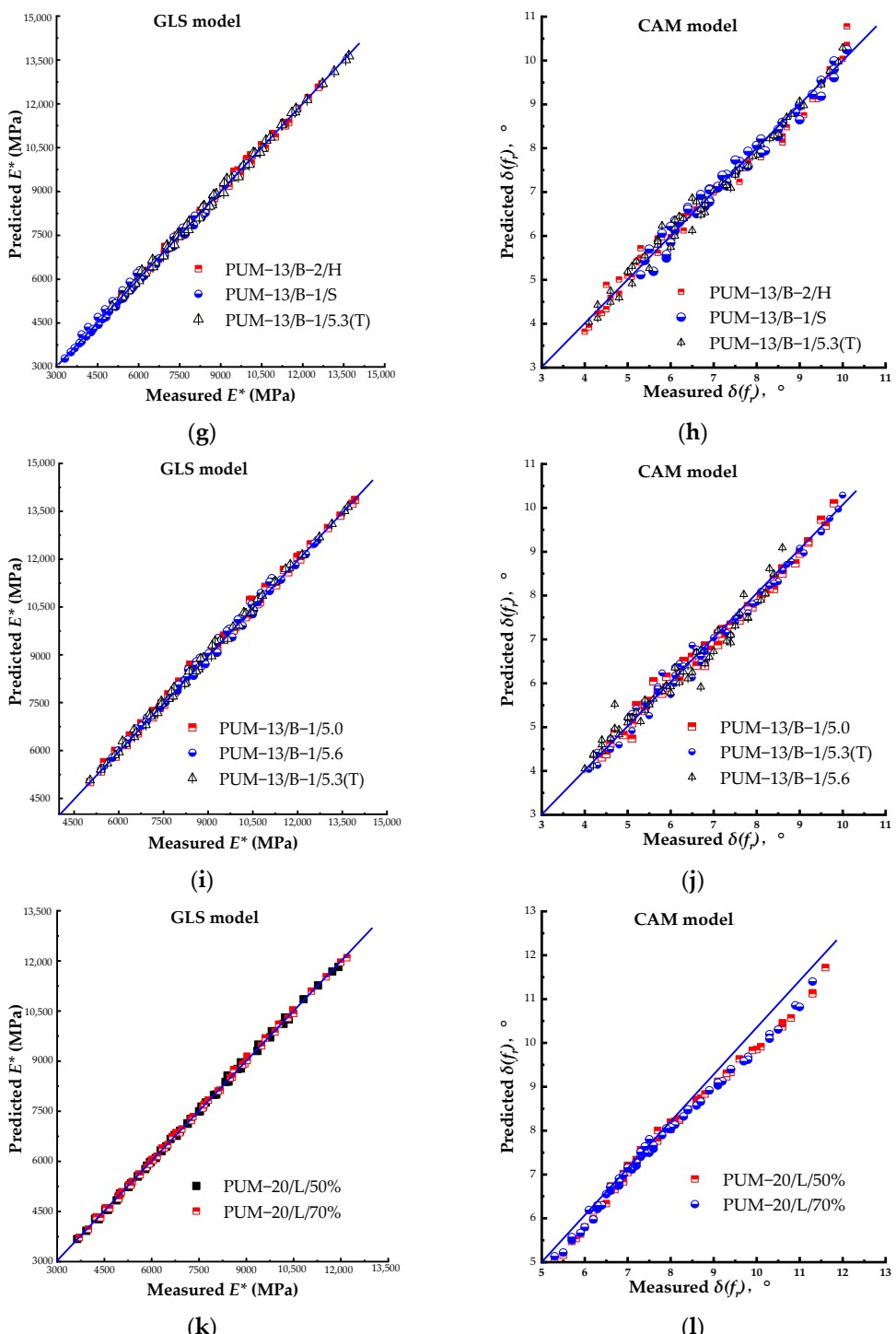

**Figure 4.** The comparison between measured and predicted dynamic modulus found by using two models. (**a**) GLS model results under different limestone aggregate gradations; (**b**) shift factors under different limestone aggregate gradations; (**c**) GLS model results under different basalt aggregate gradations; (**d**) SCM model results under different basalt aggregate gradations; (**e**) GLS model results under different aggregate types; (**f**) SCM model results under different aggregate types; (**g**) GLS model results under different aggregate types; (**h**) SCM model results under different aggregate types; (**i**) GLS model results under different PU contents; (**j**) SCM model results under different PU contents; (**k**) GLS model results under different curing conditions; (**l**) SCM model results under different curing conditions.

**Table 3.** The comparison results between measured and predicted dynamic modulus found by the GLS model.

| Type | Equation | R$^2$ |
|---|---|---|
| PUM-13/L | Y = 0.99622 × X + 35.33 | 0.99849 |
| PUM-16/L | Y = 0.99817 × X + 22.94498 | 0.99897 |
| PUM-20/L | Y = 0.9971 × X + 22.46634 | 0.99749 |
| PUM-10/B | Y = 0.99892 × X + 15.7254 | 0.99867 |
| PUM-13/B-2 | Y = 0.99751 × X + 23.82764 | 0.99891 |
| PUM-13/L | Y = 0.99622 × X + 35.33004 | 0.99849 |
| PUM-13/B-1/S | Y = 0.99511 × X + 30.09193 | 0.99482 |
| PUM-13/B-2/H | Y = 0.99759 × X + 27.40399 | 0.99806 |
| PUM-13/B-1/5.3(T) | Y = 0.99847 × X + 18.61656 | 0.99802 |
| PUM-13/B-1/5.0 | Y = 0.99553 × X + 42.56612 | 0.99761 |
| PUM-13/B-1/5.6 | Y = 0.99366 × X + 59.54671 | 0.99528 |
| PUM-20/L/50% | Y = 0.99871 × X + 15.99336 | 0.99931 |
| PUM-20/L/70% | Y = 0.99898 × X + 13.30211 | 0.9994 |

**Table 4.** The comparison results between measured and predicted dynamic modulus found by SCM model.

| Type | Equation | R$^2$ |
|---|---|---|
| PUM-13/L | Y = 0.96633 × X + 0.19561 | 0.97094 |
| PUM-16/L | Y = 0.97835 × X + 0.08778 | 0.97288 |
| PUM-20/L | Y = 0.98001 × X + 0.14631 | 0.98699 |
| PUM-10/B | Y = 0.98508 × X + 0.0764 | 0.98636 |
| PUM-13/B-2 | Y = 0.98634 × X + 0.06885 | 0.9876 |
| PUM-13/L | Y = 0.96633 × X + 0.19561 | 0.97094 |
| PUM-13/B-1/S | Y = 0.97703 × X + 0.1427 | 0.98536 |
| PUM-13/B-2/H | Y = 0.96633 × X + 0.19561 | 0.97094 |
| PUM-13/B-1/5.3(T) | Y = 0.98763 × X + 0.07962 | 0.98732 |
| PUM-13/B-1/5.0 | Y = 0.98525 × X + 0.0931 | 0.98758 |
| PUM-13/B-1/5.6 | Y = 0.9293 × X + 0.42822 | 0.95901 |
| PUM-20/L/50% | Y = 0.99932 × X − 0.00447 | 0.99122 |
| PUM-20/L/70% | Y = 0.99616 × X+0.01783 | 0.99294 |

*3.3. Analyzing the Significance of Different Influence Factors*

The ANOVA analysis method was used to ascertain the significance of various influence factors on the dynamic modulus and phase angle data, while these data were obtained from the master curves. This analysis aimed to determine whether these influence factors have a substantial effect on the dynamic modulus and phase angle of the PU mixture. The statistical analysis results are listed in Table 5.

**Table 5.** The significance of analyzing results between different groups.

| Factor | Group | *p*-Value | |
|---|---|---|---|
| | | **GLS Model** | **CAM Model** |
| | PUM-20/L & PUM-13/L | 0.056 | 0.851 |
| Aggregate gradation/limestone | PUM-20/L & PUM-16/L | 0.04 | 0 |
| | PUM-13/L & PUM-16/L | 0.891 | 0 |
| Aggregate gradation/basalt | PUM-10/B & PUM-13/B-2 | 0.004 | 0 |
| Aggregate type | PUM-13/L & PUM-13/B-2 | 0.503 | 0.077 |
| | PUM-13/B-1/5.3(T) & PUM-13/B-1/S | 0 | 0.235 |
| PU type | PUM-13/B-1/5.3(T) & PUM-13/B-2/H | 0.249 | 0.001 |
| | PUM-13/B-1/S & PUM-13/B-2/H | 0 | 0.027 |
| | PUM-13/B-1/5.3(T) & PUM-13/B-1/5.0 | 0.993 | 0.162 |
| PU content | PUM-13/B-1/5.3(T) & PUM-13/B-1/5.6 | 0.461 | 0.573 |
| | PUM-13/B-1/5.0 & PUM-13/B-1/5.6 | 0.466 | 0.403 |
| Curing condition | PUM-20/L/50% & PUM-20/L/70% | 0.71 | 0.472 |

### 3.4. Comparing Black Space Diagram

The black space diagram is the phase angle versus dynamic modulus, and is used to compare the rheology property of the asphalt mixture. Figure 5 depicts the black space diagrams of the PU mixtures with varied features.

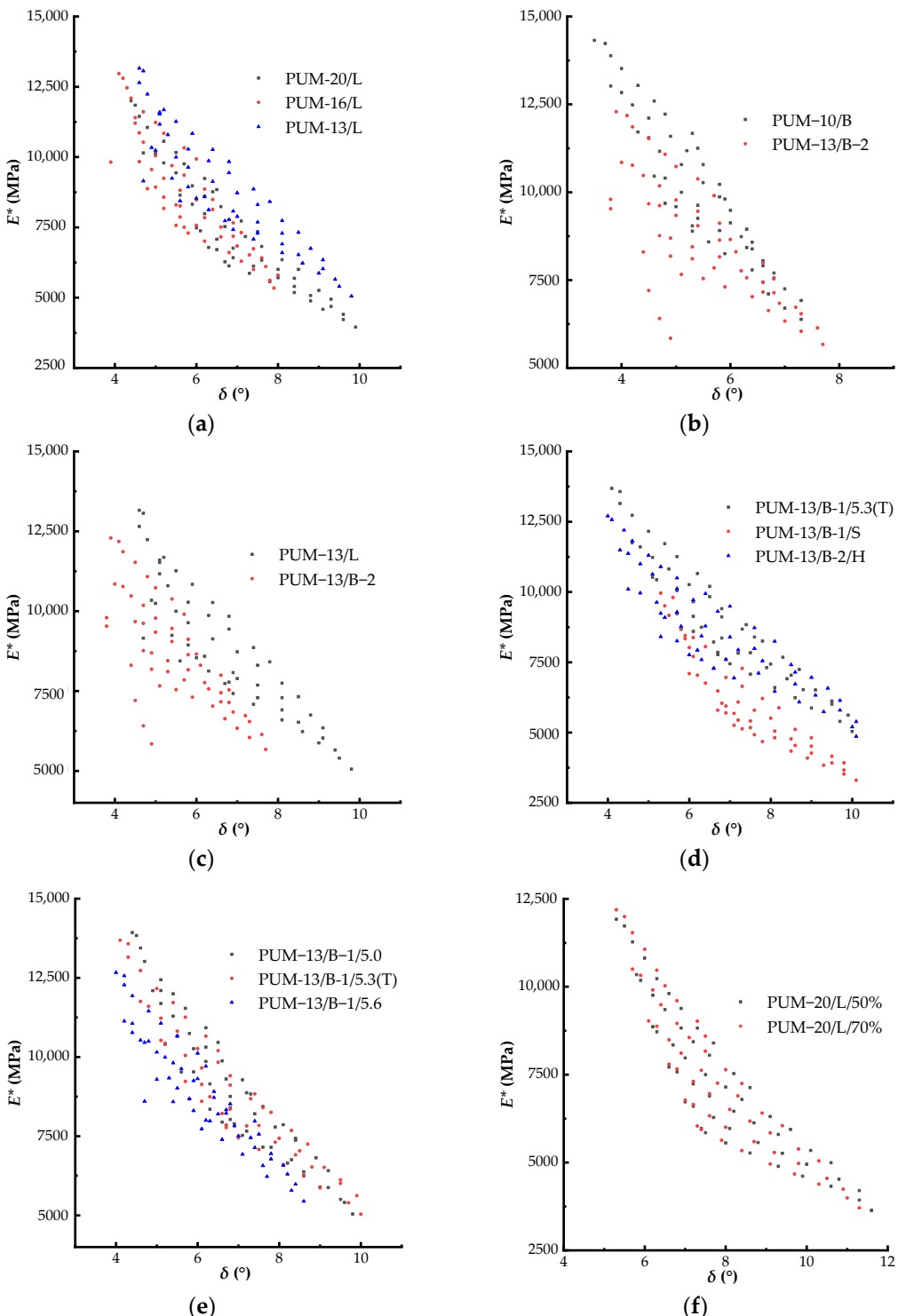

**Figure 5.** The black space diagram of phase angle versus dynamic modulus. (**a**) Different limestone aggregate gradations; (**b**) different basalt aggregate gradations; (**c**) different aggregate types; (**d**) different PU types; (**e**) under different PU contents; (**f**) master curves under different curing conditions.

## 3.5. Comparing Stiffness Parameters (E*/sin(δ))

The stiffness parameters (*E\*/sin*(δ)) are used to compare the rutting resistance of the mixture; Figure 6 represents the stiffness parameters (*E\*/sin*(δ)) of the PU mixtures with different features.

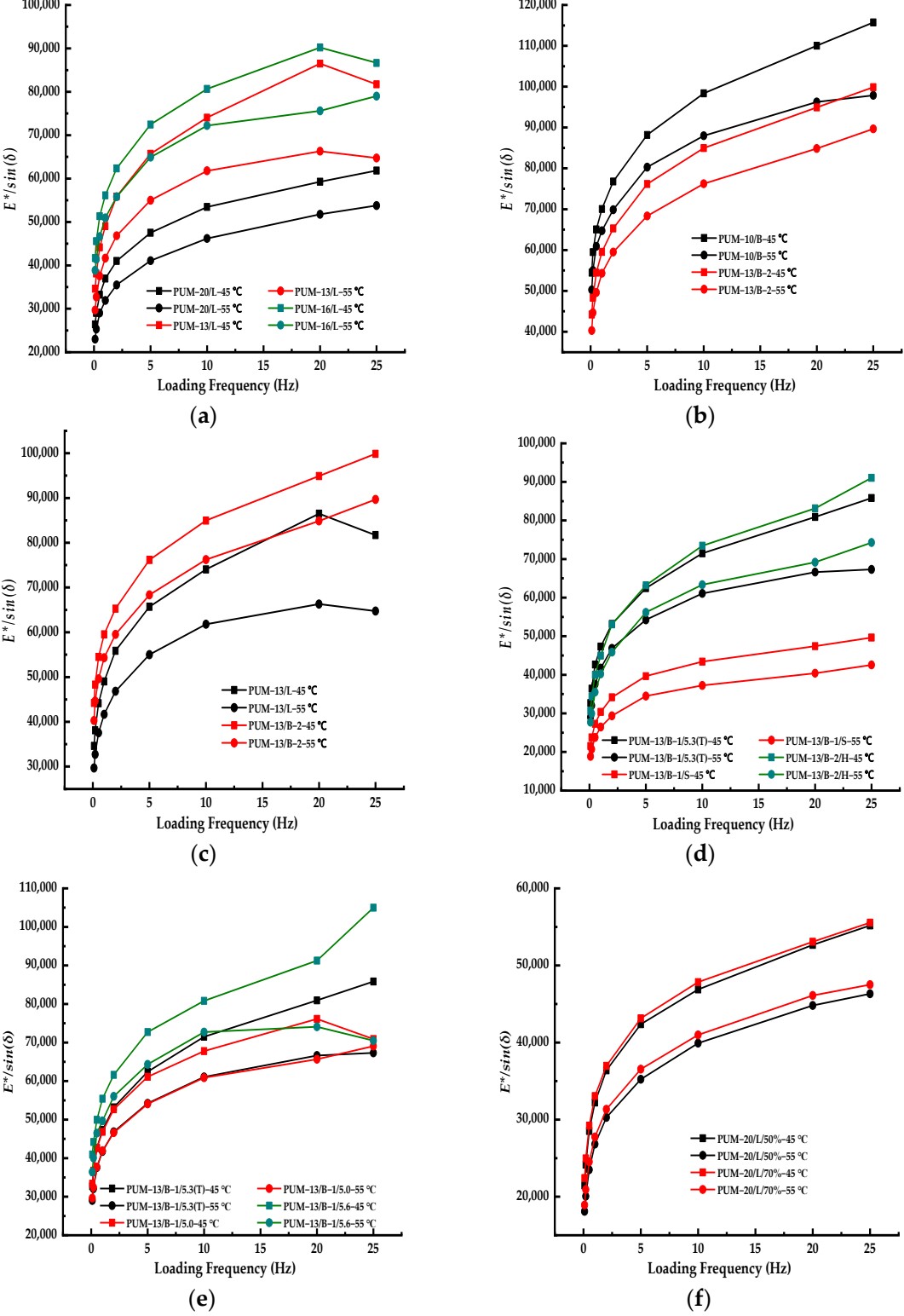

**Figure 6.** *Cont.*

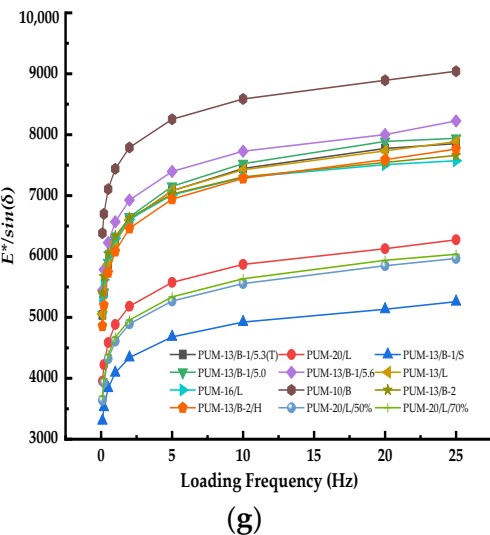

**(g)**

**Figure 6.** The stiffness parameters ($E^*/sin(\delta)$) of the PU mixture with different features. (**a**) Different limestone aggregate gradations; (**b**) different basalt aggregate gradations; (**c**) different aggregate types; (**d**) different PU types; (**e**) under different PU contents; (**f**) master curves under different curing conditions; (**g**) stiffness parameters ($E^*/sin(\delta)$) of the PU mixture with all features.

## 4. Discussion

### 4.1. Comparing Master Curve Fitting Results

#### 4.1.1. The GLS Model Fitting Results for Dynamic Modulus

Figure 2a,b demonstrate that the PU mixtures with various limestone gradations exhibited distinct dynamic modulus master curves. The master curve of PUM-20/L was lower than the other master curves. The master curve of PUM-16/L was higher than that of PUM-13/L at a lower frequency and would meet frequencies above 1 Hz. This means that PUM-20/L has a lower dynamic modulus and PUM-16/L has a higher dynamic modulus. Therefore, the nominal maximum aggregate size (NMAS) had a certain influence on the dynamic modulus, and the regularity of this influence was unclear. The order of the shift factor values or absolute values was PUM-13/L < PUM-16/L < PUM-20/L. The shift factor, which depends on temperature, is used to reflect how the master curve is translated from the dynamic modulus or phase angle curves at each temperature [45]. The temperature dependency of the mixture is given by the values of the shift factor at each temperature required to generate a master curve. The temperature dependency increases as the shift factor increases [46]. When the curve is shifted to the reference temperature, activation energy serves as the energy barrier that must be overcome. The harder it is to move the curve, the larger the activation energy. In addition, activation energy is consistent with the shift factor [47]. Therefore, PUM-20/L had the greatest temperature dependency and would require more activation energy to shift the curves at each temperature. The NMAS had a substantial effect on the construction of the dynamic modulus master curves of the PU mixture, and the dynamic modulus of the PU mixture with greater NMAS was more susceptible to temperature changes.

It is clear from Figure 2c,d that PUM-13/B has a smaller dynamic modulus master curve than PUM-10/B; therefore, the NMAS could influence the dynamic modulus of the PU mixture, but the significance of this influence was not determined. The shift factors of PUM-13/B-2 were comparable to those of PUM-10/B; both exhibited a similar temperature dependency. The influence of NMAS on the dynamic modulus master curve of the PU mixture when the NMAS was smaller than 13.2 mm needs further investigation.

Figure 2e,f illustrates that the dynamic modulus master curves of PUM-13/B-2 and PUM-13/L did not exhibit evident regularity; the dynamic modulus of PUM-13/B-2 was smaller at low frequency and higher at high frequency compared with PUM-13/L. The shift factor of PUM-13/B-2 had higher absolute values than that of PUM-13/L, but the difference

between the shift factor values at each temperature was not prominent, indicating that aggregate type had limited influence on the dynamic modulus of the PU mixture.

Figure 2g,h displays that PUM-13/B-1/S had a lower dynamic modulus master curve than PUM-13/B-1/5.3(T) and PUM-13/B-2/H. However, the master curves of PUM-13/B-1/5.3(T) and PUM-13/B-1/H were similar, which could be attributed to the fact that the color additive has little impact on the dynamic performance of the PU mixture. PUM-13/B-1/S had higher absolute shift factor values, indicating that it required more activation energy to construct the master curve and was susceptible to temperature changes. This suggests that the curing speed of the PU could affect the dynamic modulus of the PU mixture.

It can be seen from Figure 2i,j that the dynamic modulus master curves of PU mixtures with various PU contents exhibited minor differences and lacked a discernible change pattern. The shift factors also show no discernible change pattern, and the PU content did not affect the temperature dependence of the PU mixture. Consequently, the PU content showed no significant influence on the dynamic modulus of the PU mixture.

Figure 2k,l suggests that the PU mixtures cured under varied conditions (humidity condition) had nearly identical dynamic modulus master curves, indicating that the dynamic modulus of the PU mixture was not affected by the curing conditions. In addition, the shift factors under each temperature also supported this conclusion.

4.1.2. The CAM Model Fitting Results for Phase Angle

The CAM model and Kaelble shift factor equation were used to fit the phase angle data and construct the master curves of the PU mixtures; the fitting results are depicted in Figure 3. From Figure 3a,b, it is possible to conclude that the phase angle of PUM-20/L was comparable to that of PUM-13/L, and was greater than that of PUM-16/L, and the aggregate gradation or NMAS had a substantial effect on the phase angle of the PU mixture. PUM-16/L and PUM-13/L had close and smaller shift factor absolute values than PUM-20/L. This shows that PUM-20/L required more activation energy to shift the curves at each temperature to construct the master curves, and PUM-20/L would be more affected by the temperature than PUM-16/L and PUM-13/L, which have smaller NMAS. This also proves that the aggregate gradation and NMAS could affect the phase angle, although the change regularity was inconsistent with the NMAS.

It can be seen from Figure 3c,d that the phase angle of PUM-13/B-2 was greater than that of PUM-10/B, particularly at lower temperatures. When the loading frequency was higher than 10–1 Hz, the phase angle master curves exhibited a small difference. The shift factor absolute values of PUM-13/B-2 were smaller than those of PUM-10/B, indicating that PUM-10/B was more sensitive to the temperatures. Consequently, the aggregate gradation or NMAS could affect the phase angle of the PU mixture.

Figure 3e,f reveals that the phase angle of PUM-13/B-2 was smaller than that of PUM-13/L at lower frequencies and tended to be similar at higher frequencies, indicating that the aggregate type had no significant impact on the phase angle of the PU mixture. The nonregularity of shift factors between PUM-13/B-2 and PUM-13/L confirmed the previous conclusion. According to the phase angle master curves and shift factors between PUM-13/B-2 and PUM-13/L, the aggregate type had a negligible impact on the phase angle of the PU mixture.

It can be observed from Figure 3g,h that the master curve of PUM-13/B-1/5.3(T) and PUM-13/B-1/S were close to that of PUM-13/B-2/H, indicating that the curing speed of the PU binder did not affect the phase angle or dynamic property of the PU mixture. However, the color additive in the PU binder could affect the phase angle of the PU mixture. The shift factors also supported the conclusion, as the shift factors of PUM-13/B-1/5.3(T) and PUM-13/B-1/S possessed comparable values and were less than the absolute values of PUM-13/B-2/H.

From Figure 3i,j, it is apparent that the master curves of PUM-13/B-1/5.0 and PUM-13/B-1/5.6 had comparable values and trends, whereas the master curves of PUM-13/B-1/5.3(T) displayed a different trend. The shift factors also lack a discernible trend; the shift

factors of the PU mixtures with various PU contents showed different trends at different temperatures. Consequently, the PU content had no significant effect on the phase angle of the PU mixture.

It is found from Figure 3k,l that the master curves of PUM-20/L/50% and PUM-20/L/70% were nearly identical, exhibiting different values at lower loading frequencies, and the shift factors of PUM-20/L/50% and PUM-20/L/70% were nearly identical. According to the graphical analysis, the curing condition did not affect the phase angle of the PU mixture.

### 4.2. Comparing the Accuracy of Different Models

Figure 4 and Table 3 display the results of analyzing the measured and corresponding predicted dynamic modulus of the PU mixtures; these results were used to evaluate the accuracy of the master curve model. According to Figure 3 and Table 1, the data of measured and corresponding predicted dynamic modulus of the PU mixture with different features were all uniformly distributed in the line of equality (LOE), and no scattered points were observed. This phenomenon indicates that the GLS model could accurately predict the dynamic modulus of the PU mixture regardless of the mixture's features. The linear regression method was used to fit the linear relationship between the measured and corresponding predicted dynamic modulus. The fitting results in Table 1 also prove that the GLS model had high prediction precision for the PU mixture, as the line slopes of all linear fitting results were greater than 0.99 and smaller than 1, indicating that the GLS model could slightly underestimate the measured dynamic modulus of the PU mixture with high precision. The $R^2$ of the linear fitting results of all PU mixtures were greater than 0.99, demonstrating that the GLS model had a high level of prediction accuracy for the PU mixture regardless of the mixture's features.

Figure 5 and Table 4 display the results comparing and linear-fitting the measured and corresponding predicted phase angles of the PU mixtures with varied features, which could be also used to estimate the prediction accuracy of the CAM model for the phase angle of the PU mixture. According to the plots of measured and corresponding predicted phase angle, the data points were distributed along the LOE, but many points were at varying distances from the LOE, indicating that the CAM model had relatively high accuracy for predicting the phase angle of the PU mixture. This finding also corroborated the line slopes of the linear regression results, which ranged from 0.9293 to 0.99932 for the PU mixtures with varied features. The $R^2$ values of the linear regression results ranged from 0.95901 to 0.99294, confirming that the CAM model accurately predicted the phase angle of the PU mixture regardless of the mixture's features.

### 4.3. Analyzing the Significance of Different Influence Factors

ANOVA was used to analyze the significance between different groups of data. The *p*-value is the indicator of the ANOVA at a 95% significance level: if the *p*-value is less than 0.05, the analyzed groups of data have a statistical difference; in other words, the independent variable has a significant influence on the dependent variable. If the *p*-value is greater than 0.05, there is no statistical difference between the analyzed groups of data; in other words, the independent variable does not affect the dependent variable.

According to the analyzing results in Table 5, the significance level of different PU mixture groups was listed for evaluating the effect of the mixture's features on the dynamic modulus and phase angle of the PU mixture. Under the same limestone aggregate gradation, for the dynamic modulus of the PU mixture, the *p*-value of PUM-20/L versus PUM-13/L was 0.056 (greater than 0.05), indicating that the NMAS had no significant effect on the dynamic modulus master curve of the PU mixture. The *p*-value of PUM-20/L versus PUM-16/L was 0.04 (less than 0.05), indicating that the NMAS had a significant influence on the dynamic modulus master curve of the PU mixture. The *p*-value of PUM-16/L versus PUM-13/L was 0.891 (greater than 0.05), indicating that the NMAS had no significant influence on the dynamic modulus master curve of the PU mixture. In summary, the NMAS has

no discernible influence on the dynamic modulus master curve of the PU mixture. For the phase angle of the PU mixture, the *p*-value of PUM-20/L versus PUM-13/L was 0.851 (greater than 0.05), indicating that the NMAS had no significant effect on the phase angle master curve of the PU mixture. The *p*-values of PUM-20/L versus PUM-16/L and PUM-16/L versus PUM-13/L were 0.0 (less than 0.05), indicating that the NMAS had a significant effect on the phase angle master curve of the PU mixture. In summary, the NMAS had a substantial effect on the phase angle master curve of the PU mixture with continuous NMAS, and the phase angle of the PU mixture would be more affected by the NMAS than the dynamic modulus. For the PU film thicknesses of PUM-16/L (10.35 μm) and PUM-13/L (10.8 μm), which were comparable, there was no significant difference between the dynamic modulus master curve of PUM-16/L and PUM-13/L; therefore, the gradation or NMAS had no significant effect on the dynamic modulus of the PU mixture with limestone aggregate. On the contrary, the phase angle master curves of PUM-16/L and PUM-13/L differed significantly; consequently, the gradation or NMAS had a significant effect on the phase angle of the PU mixture with limestone aggregate.

The *p*-value of the dynamic modulus master curves of PUM-10/B versus PUM-13/B-2 (0.004) and the *p*-value of the phase angle master curves (0.0) were both less than 0.05 for basalt aggregate. Consequently, there were significant disparities between the dynamic modulus and phase angle master curves of PUM-10/B versus PUM-13/B-2, and the gradation or NMAS had a significant effect on the dynamic modulus and phase angle of the PU mixture with basalt aggregate.

For PUM-13/L/B versus PUM-13/B-2, which had similar gradation and different aggregate types, the *p*-values of the dynamic modulus and phase angle master curves were 0.503 and 0.077, which were both greater than 0.05. Therefore, the dynamic modulus and phase angle master curves of PUM-13/L/B versus PUM-13/B-2 had no significant difference, and the aggregate type had no significant influence on the dynamic modulus and phase angle master curves of the PU mixture. The aggregate type does not affect the dynamic property, and the PU binder is suitable for more aggregate types.

For the PU type, the *p*-value of the dynamic modulus and phase angle master curves of PUM-13/B-1/5.3(T) versus PUM-13/B-1/S were 0.0 and 0.235, respectively, indicating that the curing speed of the PU binder may alter the dynamic modulus but not the phase angle of the PU mixture. The *p*-values of the dynamic modulus and phase angle master curves of PUM-13/B-1/5.3(T) versus PUM-13/B-2/H were 0.249 and 0.001, respectively, indicating that the color additive of the PU binder can alter the phase angle but not the dynamic modulus of the PU mixture. The *p*-values of the dynamic modulus and phase angle master curves of PUM-13/B-1/S versus PUM-13/B-2/H were 0.0 and 0.027, respectively, indicating that the slow speed and curing speed of the PU binder could affect the dynamic modulus and phase angle of the PU mixture. Therefore, the PU type had a significant impact on the dynamic property of the PU mixture. This could be explained by the fact that slow curing speed or color additive could impact the ultimate morphology and strength of the PU binder. As the PU binder used in this paper is a wet-set type, the curing process of the PU binder must involve gaseous water, and different types of PU binder would require various curing periods.

All the *p*-values of the dynamic modulus and phase angle master curves of PUM-13/B-1/5.3(T) versus PUM-13/B-1/5.0, PUM-13/B-1/5.3(T) versus PUM-13/B-1/5.6, and PUM-13/B-1/5.0 versus PUM-13/B-1/5.6 were greater than 0.05, indicating that the PU content had no significant influence on the dynamic modulus and phase angle master curve of the PU mixture. The optimum PU binder content should be determined by other parameters. When the PU binder film thickness was greater than 8.7 μm for the dense gradation, the increase of the PU binder content did not improve the dynamic property.

The *p*-values of the dynamic modulus and phase angle master curves for PUM-20/L/50% versus PUM-20/L/70% were 0.71 and 0.472, respectively, which were greater than 0.05. Therefore, when the curing time is long enough, the curing condition has no significant effect on the dynamic property of the PU mixture.

*4.4. Comparing Black Space Diagram Results*

The black space diagram [48] is an effective tool for comparing and evaluating the rheological properties of asphalt binder and the performance of asphalt mixture [5,49], as well as assessing the stiffness and relaxation capability of mixtures based on a plot of dynamic modulus versus phase angle [46]. If the black space curve shifts to the right or the inflection point shift to the right, the material or the mixture would exhibit a more viscous property.

The data points in the black space plots of phase angle versus dynamic modulus cannot form a curve and are dispersed within a certain range, as shown in Figure 5. Due to the interaction of the asphalt binder with aggregate [47], the black space plot for the asphalt mixture shows a peak phase angle value at intermediate stiffness. Due to the low stiffness and viscous flow of the asphalt binder, the aggregate structure starts to dominate behavior at high temperatures, whereas at lower temperatures the asphalt mixture volumetric and binder stiffness controls the behavior. Compared with those of the asphalt mixture, the black space plots of the PU mixture with various features did not show obvious curves and peak phase angle values; this could be attributed to the PU mixture's lower temperature sensitivity.

From Figure 6a, it could be observed that the data points for PUM-13/L shift to the right, whereas the data points for PUM-16/L and PUM-20/L are mixed. This may be explained by the fact that PUM-13/L would exhibit a more viscous property than PUM-16/L and PUM-20/L, but the aggregate gradation or NMAS had a slight impact on the rheological property of the PU mixture. The data points for PUM-10/B were located on the right and in the high position, whereas the data points of PUM-13/B-2 were located on the left and in the lower position, exhibiting a more dispersed distribution than the aggregate gradation, which had a relatively small impact on the rheological property of the PU mixture. This suggests that the aggregate gradation had a negligible effect on the rheological property of the PU mixture over the test temperature range; in other words, the PU binder dominated the rheological property, whereas the aggregate skeleton played no discernible role. Therefore, the only factor influencing the rheological property of the PU mixture is the PU binder.

Figure 5c indicates that the data points of PUM-13/L were shifted to the right and some data points were mixed with those of PUM-13/B-2, indicating that the aggregate type had a certain impact on the rheological property and that the aggregate skeleton did not play a dominant role on the rheological property of the PU mixture.

Figure 5d shows that the data points for PUM-13/B-1/5.3(T) and PUM-13/B-2/H were very close and only affected by a small amount. This means that the color additives did not affect the rheological property of the PU mixture. The data points of PUM-13/B-1/S shifted to the left and displayed a more elastic property, indicating that the curing speed could influence the rheological property of the PU mixture. This phenomenon may be explained by the fact that the slow curing speed PU requires more time to set and would be in contact with more gaseous water; this process would promote the slow curing speed for the PU binder to fully set, resulting in a more elastic PU binder.

As seen in Figure 5e, the data points of PUM-13/B-1/5.3(T) and PUM-13/B-1/5.0 were closely distributed and show little difference; however, the data points of PUM-13/B-1/5.6 were slightly shifted to the lift, and part of them were mixed with the other data points of PUM-13/B-1/5.3(T) and PUM-13/B-1/5.0. This means that increasing the PU binder does not help change the rheological property of the PU mixture; in other words, when the PU binder content exceeds the optimal level, the dynamic performance is improved.

As seen in Figure 5f, the data points of PUM-20/L/50 and PUM-20/L/70% were very close and showed little difference. This could be explained by the fact that when the curing time is long enough, the ultimate performance of the PU binder is not affected by the curing condition.

*4.5. Comparing Stiffness Parameters ($E^*/sin(\delta)$)*

The NCHRP report 513 discovered that the field permanent deformation behavior has a certain relationship with the parameter of dynamic modulus. These findings were integrated into the Superpave system for evaluating pavement performance. The investigation found that the asphalt pavement's stiffness parameter ($E^*/sin(\delta)$) is accurately characterized by the asphalt mixture's rutting resistance, particularly at extensively high environment temperatures. As the stiffness parameter ($E^*/sin(\delta)$) value increased, so would the asphalt mixture's resistance to rutting, which means permanent deformation.

In this study, high temperatures of 45 and 55 °C were chosen to compare the rutting resistance of the PU mixture at high service temperatures. As shown in Figure 6, the stiffness parameter of all the PU mixtures decreases as the temperature increases. At the same temperature, the PU mixtures with distinct features would exhibit different stiffness parameter tendencies. Based on Figure 6a, the following stiffness parameters were ranked: PUM-16/L > PUM-13/L > PUM-20/L, and the stiffness parameter values of PUM-16/L and PUM-13/L were greater than those of PUM-20/L. This indicates that the gradation or NMAS substantially affects the rutting resistance of the PU mixture. According to Figure 6b, PUM-10/B's stiffness parameters were greater than PUM-13/B-2 with basalt aggregate, indicating that the PU mixture with smaller NMAS would manifest bigger stiffness parameter values or greater rutting resistance. Therefore, the PU mixture with NMAS of 16 mm for the limestone aggregate and NMAS of 10 mm for the limestone aggregate was recommended.

Figure 6c reveals that PUM-13/B-2′s stiffness parameters were greater than PUM-13/L, and PUM-13/B-2′s stiffness parameters at 45 °C were greater than PUM-13/L at 55 °C. This could be explained by the fact that the PU mixture with basalt aggregate had a greater ability to resist the deformation than the PU mixture with limestone since the PU mixtures with varying binder content had the same binder content. The aggregate was recommended for the PU mixture design.

Figure 6d shows that PUM-13/B-1/5.3(T) and PUM-13/B-2/H's stiffness parameter values were comparable and significantly bigger than PUM-13/B-1/S for the three PU mixtures with different types of PU binder that had the same binder content, aggregate type, and similar gradation. The difference in stiffness parameter values could be explained by the fact that the color additives had little impact on the rutting resistance of the PU mixture, whereas the curing speed of the PU binder had a substantial impact. The slow curing speed could reduce the PU mixture's resistance to rutting.

At 45 °C, the stiffness parameters values were ranked as follows: PUM-13/B-1/5.6 > PUM-13/B-1/5.3(T) > PUM-13/B-1/5.0; at 55 °C, the stiffness parameter lines of PUM-13/B-1/5.3(T) and PUM-13/B-1/5.0 merged and were smaller than those of PUM-13/B-1/5.6. These results indicate that the resistance of the PU mixture to rutting could be enhanced as the PU binder content increases. Because the PU binder does not produce rheological deformation at the tested temperature range, the free PU binder does not move freely as the temperatures rise, and resists rutting alongside the aggregates.

Figure 6f reveals that PUM-20/L/70%'s stiffness parameters were slightly bigger than PUM-20/L/50%'s; consequently, the curing condition had little impact on the rutting resistance of the PU mixture.

The results in Figure 6g demonstrate that at 55 °C, PUM-10/B had the greatest stiffness parameter value, followed by PUM-13/B-1/5.6, PUM-13/B-1/5.6, PUM-13/B-1/5.0, PUM-13/B-1/5.3(T), PUM-13/L, PUM-13/B-2/H, PUM-13/B-2, and PUM-16/L. PUM-20/L, PUM-20/L/70%, and PUM-20/L/50% comprised the third group with smaller stiffness parameter values, while PUM-13/B-1/S had the smallest stiffness parameter value. This indicates that PUM-10/B had the highest rutting resistance, while PUM-13/B-1/S exhibited the smallest rutting resistance. It also indicates that the curing speed had a significant impact on the rutting resistance of the PU mixture. The PU mixture with 20 mm NMAS exhibited relatively lower rutting resistance ability. The PU mixture with smaller NMAS exhibited relatively higher rutting resistance ability. These results could provide information

regarding the rutting resistance of the PU mixture and help in identifying the aggregate type and gradation of the PU mixture.

## 5. Conclusions

In this paper, six types of PU mixtures with distinct features, such as limestone aggregate gradation, basalt gradation, aggregate type, PU binder type, PU binder content, and curing condition, were analyzed and compared. The dynamic modulus test results of all the PU mixtures were compared in the form of dynamic modulus master curve, phase angle master curve, master curve fitting accuracy, the significance of influence factors, black space diagram, and stiffness parameter to determine which parameters could affect the dynamic, rheology property, and rutting resistance of the PU mixture. Based on the analysis, this paper provides information and helps in the design of the PU mixture with excellent road performance.

(1) For the limestone gradation, the gradation or NMAS influences the dynamic modulus and phase angle master curve or shift factor. The PUM-13/L data point shifted to the right and exhibited a more viscous property in the black space diagram. The stiffness parameters were ranked as PUM-16/L > PUM-13/L > PUM-20/L. Then the gradation had a substantial effect on the rutting resistance.

(2) For the basalt gradation, PUM-10/B had a larger dynamic modulus master curve and lower phase angle master curve compared to PUM-13/B-2, and the phase angle and dynamic modulus between PUM-13/B-2 and PUM-10/B exhibited significant difference. The gradation had a moderate impact on the rheological property of the PU mixture.

(3) The aggregate gradation has the potential to substantially affect the dynamic property and rutting resistance of the PU mixture.

(4) The aggregate type was proven to influence the rheological property and rutting resistance of the PU mixture.

(5) The slow curing speed reduced the dynamic modulus and resistance to rutting of the PU mixture but did not show any influence on the phase angle. The color additive did not affect the dynamic property and rutting resistance of the PU mixture.

(6) The PU binder content did not influence the dynamic modulus and phase angle of the PU mixture. The rise in PU binder content enhanced the high-temperature rutting resistance.

(7) Different curing conditions did not affect the PU mixture's dynamic property, rheological property, or resistance to rutting.

(8) The GLS and CAM models were able to accurately predict the dynamic modulus and phase angle of the PU mixture, respectively, regardless of the mixture's features.

(9) PUM-10/B and PUM-13/B-1/S exhibited the highest and lowest rutting resistance, respectively. The PU mixture with ANMS of 13 and 16 mm was the second group, and the PU mixture with NMAS of 20 mm comprised the third group.

The aggregate gradation or NMAS could affect the dynamic property, rheological property, and rutting resistance of the PU mixture; therefore, the aggregate gradation with small NMAS is recommended for the PU mixture. The aggregate type had a certain impact on the PU mixture's performance; the selection of the PU mixture should depend on its position. If the PU mixture is the surface layer, the slip resistance should be the crucial factor. The basalt aggregate should be utilized to ensure the slip performance of the surface layer. Limestone could meet the performance requirements of the PU mixture for other layers and could save investments. The curing speed of the PU binder could substantially affect the dynamic property, rheological property, and rutting resistance of the PU mixture; therefore, the slow curing speed of the PU binder is not recommended for the PU mixture. The rise in PU binder content has a mineral impact on the dynamic and rheological properties as well as the resistance to rutting. Then, the determination of the PU binder content should be investigated further through additional tests. When the curing humidity reaches 50% RH, the curing condition is attributed little to the improvement of the PU mixture.

This study analyzed and compared the factors that could influence the performance of the PU mixture, and helps in understanding the PU mixture's components and how the components affect the performance of the PU mixture. This study provides information about the PU mixture's dynamic properties and the mixture design, which will help the researchers and engineers in designing PU mixtures with excellent dynamic properties and rutting resistance.

This study was limited to six factors of the PU mixture, but more factors should be analyzed to provide more information about the property of the PU mixture. For example, more aggregate types except limestone and basalt should be researched. This study focused on the dynamic property, rheological property, and rutting resistance of the PU mixture. To improve the PU mixture design, additional tests, e.g., slip resistance and crack resistance, should be performed in further studies.

**Author Contributions:** Conceptualization, H.Z. and L.W.; methodology, S.M.; software, Z.L.; validation, H.Z. and Z.L.; formal analysis, H.Z.; investigation, H.Z. and S.C.; resources, W.Z. and P.Z.; data curation, H.Z., S.C. and P.Z.; writing—original draft preparation, H.Z.; writing—review and editing, S.M.; visualization, H.Z., S.C., C.S., Z.L. and S.W.; supervision, S.M. and L.W.; project administration, W.Z. and S.M.; funding acquisition, W.Z. All authors have read and agreed to the published version of the manuscript.

**Funding:** This research received no external funding.

**Institutional Review Board Statement:** Not applicable.

**Informed Consent Statement:** Not applicable.

**Data Availability Statement:** The data presented in this study are available on request from the corresponding author.

**Acknowledgments:** We thank Gen Li for their assistance with experiments and valuable discussion.

**Conflicts of Interest:** The authors declare no conflict of interest.

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
