# Peer review of "Study on the Influence Factors of the Dynamic Property of the Polyurethane Mixture with Dense Gradation"

_coatings, doi:10.3390/coatings13081465_

Round 1
Reviewer 1 Report
Manuscript entitles “Study On the Influence Factors of the Dynamic Property of the PU Mixture with Dense Gradation” by Zhao et al., has comments and recommends shown in below:
1. The grammatical English have to improve throughout manuscript.
2. Viscosity is one of important factor for the dynamic property of solution, why you did not study or report about this parameter. If possible, please add more the result of viscosity.
3. In the part of Introduction which relating the literature review, author mentioned some previously reported. In this case, author should put their name instead of they or the study i.e., The study [24] change to Wang et al., [24] study……..
4. What is the effect of G* on the PU mixture? You should mention it and explain why you need to study?
5. Absolutely agree with you, the rheological properties are playing a crucial role in characterizing of solution. Why you did not investigate the G* versus the strain rate?
6. In the part 2.1 Material and Gradation, what is AC−20, AC−16, AC−13, and AC−10? Should be tell more information about the symbol.
7. Author mentioned that the PU have been aggregation, right? How much size or time of aggregation? How to measurement it?
8. What is criteria for define the loading frequency and temperature for rheology test?
9. In general, the master curve would be shift higher when increased the temperature. Why author mentioned the activation energy also more increases.? It is should be decreased, right?
10. How about the G’ and G’’ of each PU mixture? All of samples exhibits same tendency?
The grammatical English have to improve throughout manuscript.
Author Response
I appreciate your suggestions and valuable advice. Based on the suggestions, I modified the manuscripts as follows.
Point 1: The grammatical English have to improve throughout manuscript.
Response 1: The English and grammatical structure were checked by English proofreading.
Point 2: Viscosity is one of important factor for the dynamic property of solution, why you did not study or report about this parameter. If possible, please add more the result of viscosity.
Response 2: The PU binder could be performed the viscosity test at room temperature before being cured, the PU binder can’t be melted into fluid status with the increasing of temperature after being cured. Therefore, the viscosity of the PU binder after being cured can’t be measured. Therefore, the viscosity of the PU binder is not suitable for analyzing the dynamic property of the PU mixture.
Point 3: In the part of Introduction which relating the literature review, author mentioned some previously reported. In this case, author should put their name instead of they or the study i.e., The study [24] change to Wang et al., [24] study……..
Response 3: The form of the literature review has been rewritten according to the advise.
Point 4: What is the effect of G* on the PU mixture? You should mention it and explain why you need to study?
Response 4: The G* is an important parameter for analyzing the stiffness and dynamic property of the PU mixture, the G* is also one of the most important input parameters for the structure design of the flexible pavement. Many researchers believed that the PU mixture could replace the asphalt mixture, it is necessary to study the G* parameter of the PU mixture and provide enough information for the pavement structure design with PU mixture layers.
Point 5: Absolutely agree with you, the rheological properties are playing a crucial role in characterizing of solution. Why you did not investigate the G* versus the strain rate?
Response 5: Thank you for the suggestion. The G* test is in strain control mode, the strain should be kept within 75-125uℇto ensure that the specimen would not produce damage, so the relationship between G* and strain rate was not studied in this paper. It is an excellent idea to study the relationship between G* and the strain rate, I will improve the test procedure to change the loading strain rate and investigate the potential relationship.
Point 6: In the part 2.1 Material and Gradation, what is AC−20, AC−16, AC−13, and AC−10? Should be tell more information about the symbol.
Response 6: The symbol AC had been rewritten in the new manuscript with a clearly expression.
Point 7: Author mentioned that the PU have been aggregation, right? How much size or time of aggregation? How to measurement it?
Response 7: I am sorry that I don’t fully under the word ’aggregation’, the word ’aggregation’ equals the curing process of the PU binder or PU mixture. The curing condition of the PU mixture was set as 35 ℃ and 50 %RH for 5 days, which was obtained from previous research. In the previous research, the specimens of the PU mixture would be kept in a chamber under different temperatures and humidity for different times, after the curing process, the specimens under different curing conditions would be subjected to a splitting test to measure the splitting strength. The recommended curing condition was based on splitting strength results.
Point 8: What is criteria for define the loading frequency and temperature for rheology test?
Response 8: The selected loading frequency and temperature would attempt to simulate the real temperature and vehicle speed of the road pavement. The highest and lowest temperatures are defined by the capability of the test chamber. The loading frequency would simulate the vehicle speed at different levels. There were no explicit criteria for defining the loading frequency and temperature. All the selected loading frequencies and temperatures are based on the literature or other research.
Point 9: In general, the master curve would be shift higher when increased the temperature. Why author mentioned the activation energy also more increases.? It is should be decreased, right?
Response 9: The activation energy corresponds to the shift factor. The bigger values of the shift factor, the more activation energy would be needed to shift the master curves to the reference temperature.
Point 10: How about the G’ and G’’ of each PU mixture? All of samples exhibits same tendency?
Response 10: The G’ and G’’ of each PU mixture would follow a similar tendency under different conditions. Therefore, the G’ and G’’ was not involved in this paper.
Reviewer 2 Report
Do not use any abbreviation in Titles?? PU and check others too
Conclusion section is too long, split the section into Conclusions, Recommendations from the study, Limitations and scope for future work
The difference between your existing studies and the proposed work is not clear, how this work is novel and what the author will gain out of this research?
“Study on the Influence Factors of Dynamic Modulus and Phase Angle of Dense Gradation Polyurethane Mixture”
Not able to see the experimental pictures to support the results?
The specific needs of utilization of basalt aggregate and PU to be explained?
The discussion section can have some exiting work and its typical comparison
The need of study and application should be more explained in introduction
Author Response
I appreciate your suggestions and valuable advice. Based on the suggestions, I modified the manuscripts as follows.
Point 1: Do not use any abbreviation in Titles?? PU and check others too
Response 1: The abbreviation of PU in the title had been changed, and other abbreviations were checked.
Point 2: Conclusion section is too long, split the section into Conclusions, Recommendations from the study, Limitations and scope for future work
Response 2: The conclusion section had been improved according to the advice.
Point 3: The difference between your existing studies and the proposed work is not clear, how this work is novel and what the author will gain out of this research?
Response 3: The limitation of the study and proposed work had been checked and rewritten. This study could help the researcher and engineer in understanding how the components of the PU mixture could affect or improve the dynamic-related property and performance of the PU mixture.
Point 4: Not able to see the experimental pictures to support the results?
Response 4: The framework of the template requires that the test results and pictures were shown in section 3, and the corresponding results discussions were shown in section 4. This form would confuse reading the paper. I am sorry about that. But all the results were supported by corresponding experimental pictures.
Point 5: The specific needs of utilization of basalt aggregate and PU to be explained?
Response 5: The explanation about the aggregate type selection was improved in the new manuscript.
Point 6: The discussion section can have some exiting work and its typical comparison
Response 6: Exiting literature was mostly focused on the performance improvement of the PU mixture compared to the traditional asphalt mixture. The conclusion in the asphalt mixture was quite different from that of the PU mixture. Therefore, this paper didn’t provide the typical comparison in the discussion section.
Point 7: The need of study and application should be more explained in introduction
Response 7: The need for study and application was improved in the introduction of the new manuscript.
Reviewer 3 Report
The manuscript is publishable at this stage
Moderate editing is required.
Round 2
Reviewer 1 Report
-
Reviewer 2 Report
The revisions are satisfactory